# Experimental Investigations upon Ultrasound Influence on Calefaction of AdBlue in Selective Catalytic Reduction Systems (SCR)

**DOI:** 10.3390/mi14081488

**Published:** 2023-07-25

**Authors:** Claudiu Marian Picus, Ioan Mihai, Cornel Suciu

**Affiliations:** Faculty of Mechanical Engineering, Automotive and Robotics, Stefan cel Mare University, 720229 Suceava, Romania; claudiu.picus@usm.ro

**Keywords:** AdBlue, heat transfer, crystallization, SCR, droplets, microchannels

## Abstract

The present paper intends to provide an analysis of how the process of calefaction occurs in a selective catalytic reduction (SCR) system and the mechanisms by which the deposition of AdBlue crystals on a hot surface evolve. Experimentally, two aluminium samples heated to 200 °C were used, over which AdBlue droplets with different atomisation rates were dropped, maintaining the same dynamic flow parameters, in order to observe the influence of temperature effects on the degree of deposition of crystallised sediment on the surface. The authors proposed the use of calefaction in an ultrasonic environment to prevent deposition and to increase droplet fragmentation by a break-up process. To prove the performance of this method one sample was subjected to a normal flow regime while a second sample was exposed to ultrasound. Both samples were assembled on a magneto-strictive concentrator operating at a frequency of 20 kHz. The obtained results indicated that the sample exposed to ultrasound demonstrated lower urea crystallisation compared to the sample that was not exposed to this treatment. Thus, it can be seen that the proposed method of injecting AdBlue into an ultrasonic zone gives the desired results.

## 1. Introduction

In the current situation of global challenges associated with air pollution and exhaust emissions from trucks and marine shipping with very heavy-duty diesel engines, SCR technology and the use of the additive AdBlue have become increasingly important for reducing nitrogen oxide (NO_x_) emissions. AdBlue is now commonly used as an emission reducing agent in the SCR process, where it reacts with nitrogen oxides to convert NO_x_ to nitrogen and water. However, the improvement of the efficiency and performance of SCR systems for cars, trucks, and marine ships remains a priority. One of the most important issues in this area is the efficient heating and vaporisation of AdBlue before entering the catalyst, especially during the time between engine warm-up and operating temperature. In modern configurations, AdBlue is injected into the exhaust manifolds of diesel engines above SCR systems to remove pollutants and neutralise NO_x_. We proposed to study how the calefaction process takes place. It consists of an intense vaporisation of AdBlue droplets on the surface of a liquid located near a heated solid body, in this case the intake manifold of a combustion engine. Vaporisation takes place throughout the volume and on surfaces, the influence of vaporisation in the ultrasonic field is studied in this article. AdBlue is composed of 32.5% urea and 67.5% distilled water, so when the water evaporates, the urea tends to crystallise and causes deposits on the exhaust system walls and at the inlet of the SCR microchannels, hindering their correct operation and performance. Many authors [1,2,3,4,5,6] have studied the crystallization of urea from a droplet of evaporating aqueous solution placed on a heated flat surface, observing macroscopically the morphology of crystal growth under different conditions. It was seen that, after a certain period of time, increasing concentrations of urea in the droplet caused crystal growth. Moreover, thermal imaging indicated an exothermic character of the process through the significant release of heat on the surface where solidification occurred. Other researchers [7,8,9,10,11], have studied methods to prevent crystallization with constructive techniques such as gas mixed with AdBlue solution in order to improve gas atomized liquid droplet homogenization and to increase the performance of the selective catalytic reduction (SCR) system. Several studies can be found in the literature that investigate the use of ultrasonic vibrations in catalytic reactions, with numerous applications in green energy production [12,13,14,15,16]. In this context, the present work intended to find solutions to reduce urea crystallization especially in the microchannels of the SCR system using an ultrasonic process. The effect of vibrations on crystal deposition during evaporation of atomized AdBlue droplets in high temperature gaseous media and in direct contact with a heated disc was also considered during the presented investigations. The mechanism of the crystallisation process and the microscopic deposition of the formed crystal, in the presence and absence of vibrations, were examined in particular. The characteristics of the crystals formed, and their rate of formation depend on the working temperature of the gaseous media, which can be controlled by efficient preheating methods in the immediate period after the engine is started up to operating temperature. The use of this method helps to avoid residue deposition on SCR system components, which has a direct impact on SCR catalyst performance and durability.

## 2. Determination of the Temperature Distribution of the Ultrasonic Generator Disc and the Droplets on Its Surface

However, as discussed, vaporisation of AdBlue droplets at a high temperature surface can lead to the appearance and formation of crystal deposits. These deposits on the inlet face of the SCR catalyst can block or even obstruct the inlets of the SCR microchannels; in this case it will lose its exhaust gas cleaning capacity. This paper presents a method to control and reduce the deposits using the effect of an ultrasonic generator which produces vibrations through a magneto-strictive concentrator to a disc. AdBlue droplets are injected on the disc. The vibrations of the disc provide a much more accentuated dispersion of the droplets already atomised by injection, with a direct effect on the deposits. The method ensures a significant quantitative reduction of solid deposits when combined with a precise temperature control of the gaseous medium. The vibrations produced have a fundamental role in improving heat transfer within the droplets, but also on mass transfer.

### 2.1. A Description of the Processes That Take Place When Injecting Adblue onto a Disc with and without Vibration

A two-phase solution is considered in this case, where the turbulence created by vibrations in the boundary layer and in the bulk result in a decrease in thermal resistance, amplifying the rate of heat transfer. The equation used in the case of solid-fluid particles for a vibrating disc is given by Strouhal’s homo-iconicity criterion, denoted *Sh*:(1)$$Sh=0.746Re^{0.5}Sc^{0.33}{\left(\frac{A}{r}\right)}^{1/6},$$

In Equation (1) the following notations are made: Reynolds criterion (*Re* = 2*ωAr*/*ν_f_*), *ω*—the pulsation, *A*—the amplitude, *r*—radius, *ν_f_*—the vibration frequency, *Sc*—the Schmidt criterion (*Sc* = *η*/*ρD*). Equation (1) was used to calculate the temperature in the droplet as a function of the distance from the centre of the droplet. The temperature is assumed to change according to an exponential function and an attenuation factor given by exp(−*rs*^2^) where, *s*—distance from the centre of the droplet. The larger the s, the smaller is the attenuation factor, while the temperature at the initial distance (denoted *T_p_*) decreases progressively. The temperature inside the droplet can be calculated with the following relationship:(2)$$T_{d}=T_{i}+(T_{p}-T_{i})exp(-rs^{2}),$$
where: *T_d_*—represents the temperature at a certain distance s inside the droplet, *T_i_*—represents the initial temperature in the droplet, *T_p_*—represents the temperature at the initial distance *s* = 0 (which can be considered as the temperature of the disc), *r*—is the radius of the droplet, *s*—is the distance from the centre of the droplet where the temperature *T_d_* is measured. In order to have a more accurate view of the vaporization phenomenon, the finite conductivity model was considered, which does not consider the study of phenomena occurring inside the droplet, but rather the thermal diffusion in the gas [17]. Convection in the droplet is neglected and concentration and temperature gradients are taken into account [18,19]. The temperature evolution is represented using the energy conservation equation below:(3)$$\frac{\partial T}{\partial t}=\frac{\lambda_{i}}{\rho C_{p}r^{2}}\frac{\partial }{\partial r}\left(r^{2}\frac{\partial T}{\partial r}\right),$$
where: *T*—temperature, *t*—time, *λ_i_*—heat conduction coefficient, *ρ*—density, *C_p_*—specific heat at constant pressure, *r*—radius. In order to analyse the evolution of the exothermic effect of the droplet, the parameters in Table 1 were used by considering the temperature variation under conditions of whether the disc is vibrated or not.

### 2.2. A Description of the Processes That Take Place When Injecting AdBlue onto a Disc with and without Vibration

The evolution of disc temperature *T_disc_* at time *t* on the surface of the disc was determined for two cases, that is without and in the presence of ultrasound [17]. The equation used for the first case is as follows:(4)$$\;T_{disc}=T_{i}+\left(T_{p}-T_{i}\right)\left(1-exp(-t)\right).$$

The mathematical calculation of the temperature variation over time when using the magneto-strictive concentrator for generating vibration is $T_{dv}$ below:(5)$$\;T_{dv}=T_{i}+(T_{p}-T_{i})(1-exp(-t)+A\sin(\omega t)),$$
where: *ω*—is the pulsation expressed by the relation *ω* = 2π*f*, *f*—frequency. The calculation results obtained in MATLAB R11b are those in Figure 1a,b, considering a transient regime where at *t* = 0 the disc temperature is that of the ambient.

Figure 1a,b shows that there is a temperature difference for the two cases. In contrast to the first case (without vibrations Figure 1a), we remark that in Figure 1b, the field marked with yellow indicates an increase in the temperature Δ*T*, resulting from vibrations of frequency 20 kHz. In the analysis of the 3D temperature distribution of the heated disc, the temperature of the disc was calculated for each point on the radius, knowing the values of the heat flux *Φ*, with the following relationship:(6)$$\;T_{disc}=T_{di}+\frac{\Phi }{4k_{d}}{\left(\frac{1-r^{2}}{2}\right)}^{2},$$
where: *T_di_*—initial disc temperature, *k_d_*—disc conductivity, *r*—disc radius.

The calculation results obtained in a MATLAB code are shown in Figure 2.

When the disc reaches the maximum temperature and the AdBlue drops are injected and impinge on the disc surface, a heat transfer takes place. Thus, in the case of a sudden surface energy pulse, *E* = lim(qst) [kJ/m^2^], and as time t approaches zero, at the surface the instantaneous temperature response is given by Equation (7):(7)$$\;T_{m}-T_{0}=\frac{E}{\rho C_{p}\sqrt{\pi at}}exp\left(\frac{x^{2}}{4at}\right),$$
where: *T_m_*—temperature at the calculated moment, *T*_0_—initial temperature, *E*—surface energy, ρ—density, *C_p_*—specific heat at constant pressure, *a*—coefficient of thermal diffusivity, *t*—time, *x*—heat propagation distance. Considering the initial temperature of the disc as 200 °C and considering the thermal inertia when the droplet contacts the hot surface of the disc, a cooling of the disc of about 34 °C occurs. Figure 3 represents the temperature change on the radius of the disc if the liquid droplet remains on the surface for 1 to 20 ms if no vibrations are applied.

The calculations show that after 1 ms the disc temperature in the centre of the droplet has the lowest value 165 °C which can be explained by the large amount of liquid in the centre of the droplet. With increasing distance x from the centre of the droplet to the extremity of the droplet, the disc temperature increases to the maximum value of 200 °C, as the amount of heated liquid decreases progressively. At the centre of the droplet when *x* = 0, after 5 ms the temperature in the centre of the droplet reaches 184 °C, after 10 ms it reaches 188 °C, and at 20 ms it reaches 192 °C. On the other hand, at the droplet tip (*x* = 3.7 mm) temperatures at the calculated moment are close to the initial disc temperature, their values being between 198 °C and 200 °C. For the case where vibrations are applied to the disc, the following equation was used:(8)$$\;T_{m}-T_{0}=\frac{E}{\rho C_{p}\sqrt{\pi at}}\exp\left(\frac{2x{\left(j\right)}^{2}}{4at\sin(2\pi ft)}\right),$$

The results obtained in a Matlab code are shown in Figure 4.

In Figure 4 we see that the blue line corresponding to the first millisecond of contact between the AdBlue droplet and the vibrating disc represents the case of a pulsating process. Compared to the previous variant, where the droplet–disc contact was all over the surface, in this case the contact times are much shorter. In Figure 4 it can be seen, due to the vibrations, how a much more pronounced heat transfer takes place. In contrast to the previous case, even if we move away from the centre of the droplet, due to the phenomenon of enhanced diffusive heat transfer, the cooling of the disc is more pronounced in the centre of the droplet, which is explained by the mass transfer. On the other hand, the heat transfer is more intense than in the previous case at the periphery of the droplet, which can be explained by the pronounced increase in molecular agitation.

### 2.3. Determination of the Evaporation Rate of AdBlue Droplets and Modelling This Phenomenon in ANSYS

The evaporation rate of atomized droplets on a high temperature quasi-isothermal cylindrical surface is calculated using the equations below. Analysis of the results obtained, allows a better understanding of the behaviour of AdBlue droplets during vaporization under the specific conditions of the experiment.

Equation (9) represents the variation of diameter of the droplet as a function of media temperature and time:(9)$$D_{i}=D\exp\left(\frac{2m_{d}iT_{a}t{\left(j\right)}^{2}}{D^{2}}\right),$$

Equation (10) represents the temperature variation on the disc surface:(10)$$\;T_{disc}=T_{di}+\alpha (T_{a}-T_{p})exp(-T(j)),$$

Considering Equations (8) and (9) it can be written that the evaporation rate *E_r_* is according to Equation (11):(11)$$\;E_{r}(j)=\left(\frac{m_{d}}{m_{d}+\alpha }\right)\pi {D_{i}}^{2}C(T_{a}-T_{p})\left(\frac{1+A}{1000}\right),$$
where: *E_r_*—evaporation rate, *m_d_*—initial mass of the droplet, *α*—mass transfer coefficient, *D_i_*—initial diameter of the AdBlue droplet, *C*—AdBlue concentration, *T_a_*—ambient temperature, *T_disc_*—disc temperature, *A*—oscillation amplitude.

Figure 5a,b presents the droplet evaporation rate as a function of time at different ambient temperatures, without the use of vibrations.

Analysing Figure 5a,b it can be seen that the evaporation rate increases with the ambient temperature, Ta, for both variants. If the disc is not subjected to vibrations, a very low evaporation rate is observed at 50 °C, with a value of about 1 g/s, and a maximum value at 500 °C not exceeding 20 g/s. In the case of the theoretical presence of vibrations on the test disc obtained by means of a magnetostrictive concentrator, calculations using a Matlab code obtained the following in Figure 6a,b.

In the situation where vibrations are present, as shown in Figure 6a,b, the evaporation rate is 5 g/s at a temperature of 50 °C, indicating slow evaporation (blue line, Figure 6a, and at a maximum temperature of 500 °C, the evaporation rate changes rapidly to 100 g/s.

Analysis of the figures shows that the evaporation rate increases with the ambient temperature in both cases and that vibrations can significantly influence this parameter. It was found that the highest evaporation rate is obtained at the beginning of atomization, this decreasing with time.

In to establish how droplet atomization proceeds on a high temperature surface and to visualize the temperature evolution of AdBlue micro-droplets, a simulation was performed in the ANSYS programming environment. This simulation considered droplets in contact with a surface subjected to vibrations generated by the magneto-strictive concentrator. The result of the simulation in Figure 7 provided a prediction of the temperature evolution and dynamics of the AdBlue micro-droplets during interaction with the magneto-strictive concentrator.

During the vaporization phenomenon induced by the vibrations generated by the magnetostrictive concentrator in contact with the AdBlue droplets, a deep crumbling of the already atomized droplets is observed due to the intensification of thermal and mass diffusion in the system. In Figure 7 the prediction of the dispersion and temperature evolution of AdBlue droplets in a controlled environment was performed for a distance Δ*x* of 10 mm from the vibrating disc. A pronounced heat transfer is observed in the 10 mm area, and subsequently the particles reach the ambient temperature relatively quickly. On the right side of the figure an experiment performed by the authors is shown for AdBlue droplets diffused into the air using an ultrasound generator, illustrating the closeness of the results with those from ANSYS.

## 3. Experimental Results and Interpretation

### 3.1. Description of the Experimental Laboratory Test Bench

In order to investigate the evaporation process and the evolution of deposits resulting from the evaporation of water from the AdBlue solution, laboratory tests were carried out on droplets in contact with a hot surface with or without using ultrasound. The experiments focused on the impact of ultrasonic vibrations on the microscopic deposition of AdBlue crystals on two disc-shaped samples. Figure 8 shows the equipment used for the experiments.

The significance of the notations in Figure 8 is as follows: 1—frequency generator, 2—AdBlue drop (illustrative picture), 3—disc-shaped samples, 4—magnetostrictive concentrator, 5—stalagmometer, 6—hot air blower, 7—thermometers. The AdBlue droplet 2 flow is controlled by the stalagmometer 5 over the surface of the disc-shaped samples 3. The disc-shaped samples 3 are rigidly attached to the tip of the magnetostrictive concentrator 4 which generates vibrations due to the frequency induced by the frequency generator 1 which can generate 20 kHz. The AdBlue droplet 2 flows into the air perpendicular to the disc 4 samples followed by diffusion with the hot air flow at 200 °C generated by the blower 6. The heating to any temperature of the samples 3 and hence the ambient environment in the area is controlled by temperature sensors via thermometers 7. The results of these tests can be used to better understand the calefaction phenomenon between a heated flat surface and AdBlue droplets in order to optimise the efficiency of the catalytic emission reduction process.

The process that takes place during the conducted experiments is illustrated graphically in the schematic diagram shown in Figure 9.

Stage 1 of the process starts with the detachment of an AdBlue droplet from the stalgmometer. First, the droplet falls perpendicularly towards the disc surface, as its dynamics is only influenced by gravity. In the second portion of the trajectory, the droplet follows a slightly curved path because its trajectory is influenced by both gravity and the heated air flow. Once the droplet reaches the heated surface, it suffers a “break-up” phenomenon and fragments into multiple smaller droplets.

Stage 2 of the process consists of the intensification of the fragmentation phenomenon under the influence of ultrasonic vibrations. This leads to obtaining atomized spherical AdBlue nanodroplets that are spread over the whole surface of the disc, thus facilitating the vaporisation process and the transfer of mass and heat.

### 3.2. Use of High-Speed Recording to Highlight the Evolution of AdBlue Droplets in the Calefaction Process

For detailed analysis of the results, fast 240 fps shooting was performed using trouble shooter equipment produced by Fastec Imaging Corporation Model TSHRCS. The disc of element 4 in the figure is heated to a temperature of 200 °C by means of a stream of hot air. Two samples are used, one of which is subjected to vibrations generated by a magnetostrictive concentrator. After the experiments, the physical properties and structure of the AdBlue crystals are evaluated with and without vibrations. This research contributes to the understanding of the crystallisation process of AdBlue under controlled vibration conditions and provides new insight into improving the efficiency of SCR systems for cars, trucks, industries, and marine ships. For this study, aspects such as air flow velocity, pressure, ambient temperature, and geometrical characteristics of the disc on the magnetostrictive concentrator were considered. Using computational analysis methods and appropriate mathematical models, results were obtained that provided information on the temperature distribution and heat transfer in the system under analysis. Regarding the thermal aspect, an aerotherm was used to heat the vibrating disc and the environment, which provided heat to both the air and the vibrating disc at 200 °C. Then, a droplet of AdBlue having a temperature of 25 °C and a mass of 100 mg was allowed to flow through a capillary tube until the droplet had direct contact with the surface of the hot disc. By heating the disc and the adjacent space, a favourable environment was created for rapid vaporisation of the water in the solution, causing residual deposits to be formed. During the experiments, important dynamic parameters such as temperature, mass and evaporation rate were monitored and recorded. This information was collected in order to obtain accurate and relevant data, allowing a detailed analysis of the behaviour of the AdBlue droplet and its interaction with the hot disc. Figure 10 shows the results of a high-speed recording to analyse the heating process of a droplet dropped in the direction of the red arrow on the high temperature disc. The vaporization process takes place instantaneously, with the droplet jumping up and down from the surface due to the breaking of contact.

In this stage of the experiment (Figure 9), the disc is not subjected to vibrations, in which case the “break-up” phenomenon of the droplet can be observed when it comes into contact with the heated surface. This “break-up” phenomenon refers to the dispersion of the droplet into smaller fragments due to impact with the hot surface. In the second phase of the experiment, the disc is subjected to vibrations, and thus a much stronger impact on the vaporisation and break-up phenomenon occurs. The calefactory mechanism is totally different in this case, the vibrations induced by the magnetorestrictive generator intensifying the break-up process. Figure 11 shows that the mass and temperature transfer in the case of vaporisation is significantly higher compared to the previous situation where the droplet is not subjected to vibrations.

On contact with the surface of the strongly heated disc, a certain part of the droplet does not vaporise instantly but breaks up into relatively large droplets. At this stage the vibrations help to disperse the droplet into nanodroplets and subsequently spread them over the surface of the disc, thus facilitating the vaporisation process and the transfer of mass and heat.

### 3.3. Application of the IR Technique to the Calefaction of AdBlue Droplets

In addition, an infrared (IR) camera was used to better highlight the differences mentioned between the use and non-use of ultrasound, and the results obtained are shown in Figure 12 and Figure 13. This technique allows thermal visualization of the process, which helps to distinguish the degree of evaporation and vaporization for the same quantity of AdBlue. With the IR camera, it is possible to observe how the heat is dispersed in the droplets as well as how the vaporisation process develops.

The figures show the totally different evolution of the AdBlue droplet break-up phenomenon, showing that the use of a magneto-restrictive concentrator produces fragmentation thousands of times amplified compared to the normal injection case.

### 3.4. Evidence of Crystal Deposition in the Case of Calefaction Phenomenon

The crystallised deposits which result from thermal decomposition of AdBlue aqueous solution, can have a negative impact on the efficient operation of selective catalytic reduction (SCR) systems. The appearance of salts through the calefaction process can lead to clogging of part of the SCR microchannels while also partially obstructing the flow of flue gas through the system. Deposition can lead to increased emission of NO_x_. In this context, the study of the evaporation and vaporization of the AdBlue droplet revealed the formation of deposits in different amounts for the two samples, as illustrated in Figure 14 where the colour images were obtained with an IR camera code FLIR TG165. These deposits can interfere with the gas and chemical flow of SCR installations and affect the overall performance of the system.

Investigations revealed an intensification of mass transfer processes in the presence of vibrations in AdBlue droplets, leading to a reduction of deposits on the sample surface. In the case of the evaporation phenomenon, the deposition of solid urea creates a rough region, marked in red in Figure 14. The liquid film tends to move towards the yellow marked area during evaporation. Furthermore, experiments showed that the deposition rate in the areas subjected to vibration was much lower than in samples that were not influenced by vibration, as shown in Figure 14b. To evaluate the deposition of AdBlue salts, measurements of the resulting surface microtopography were performed using a MarSurf CWM 100 confocal microscope and interferometer produced by Mahr GmbH, Gottingen, Germany (Figure 15).

Measurements of the surface microtopography were made in the central region of the samples, in 2 × 1.5 mm^2^ areas, considering both longitudinal and transverse directions. In order to quantitatively determine crystal deposition on the surfaces of the two samples (with or without vibration), quantitative analyses were performed at selected points of interest on each piece. These points were placed from right to left on the part surfaces using a random sampling method. The areas of interest, illustrated in Figure 16, were identified based on specific criteria, such as roughness level and deposit distribution, in order to obtain a more precise assessment of the amount and distribution of deposits on the surfaces.

The analysis of these areas of interest involved the use of high-resolution microscopy with a Mahr CWM 100 confocal microscope, using specific analytical methods to process the results. The use of the confocal microscope, allowed a detailed analysis of the height of the deposits for the two samples investigated, as shown in Figure 17a,b.

From the microscope measurement reports, it can be seen that there are significant differences between the two cases in terms of the height of the deposition profile. In case (a), where no ultrasound was used, the height of the deposition profile is much higher compared to case (b), where ultrasound was used. This suggests that the use of ultrasound has a positive effect in reducing the height of the deposits, thus contributing to the partial or total removal of the crystals formed by calefaction. Where no vibration was used, it can be seen from Figure 18 that a significant accumulation of solid deposits occurs, with heights of the order of 370 μm.

These deposits are more concentrated in certain areas of the analysed surface compared to the case where vibrations were used. The distribution of residual solid particles does not cover the entire studied surface to the same extent as when using vibration. These crystals accumulate in larger concentrated areas, which can create significant variations in deposition between different regions of the surface. Analysing the profile of the samples in the case of vibration shown in Figure 19, a constant distribution of asperity peaks can be observed over approximately 80% of the surface.

Small clusters of crystals are also found concentrated in certain areas of the surface, where differences of up to 50 μm can be observed. These differences represent a significant variation in heights in those areas and can have a significant impact on the remaining deposition. Using confocal microscopy, it was possible to determine the roughness parameters of the profiles, Ra and Rz, Figure 20.

These parameters are essential in evaluating surface roughness properties. For the analysed profile, an average roughness value Ra of 16.043 μm was obtained for the case where there were vibrations, respectively, 31.122 μm for the sample that was not subjected to vibrations. Ra roughness values are important in surface characterization because they reflect the absolute heights of the elevations and depressions on the surface. Thus, lower values of Ra indicate better surface quality with reduced roughness and more uniform texture.

## 4. Conclusions

The present article discusses the importance of eliminating all or part of the urea formations and deposits on the microchannels of SCR systems. The practice of injecting AdBlue into SCR systems is commonly used and is still being improved upon. One method is to heat the exhaust gases in the engine warm-up mode until the operating temperature is reached. During this period the emissions are much higher. The present paper showed that injecting AdBlue into the gas stream at high temperatures can cause deposits to form through calefaction under specific conditions. These deposits are undesirable in the operation of SCR systems, so the authors propose the following two routes:injecting the AdBlue solution into an ultrasonic fieldgas temperature control with an external induction heating system (not part of this study).

The experiments carried out and the results obtained have undoubtedly shown that major differences in the heating phenomenon occur in the presence or absence of a vibration-generating ultrasonic field. The images obtained in Figure 10 and Figure 12 without vibrations and Figure 11 and Figure 13 with vibrations show clearly how the AdBlue droplets go through an advanced process of fragmentation through a break-up effect; in such a case the vaporization process will also be much more efficient. This increase in vaporization intensity is attributed to the increased convective surface area of the atomized spherical AdBlue droplets and the increased velocity difference of the ultrasound relative to the gaseous medium in which it travels. In our opinion, with minimal investment, the atomization and reduction of NO_x_ when injecting AdBlue into a gaseous medium gives satisfactory results. In addition, even in the case of a calefaction process, AdBlue droplets in quasi-periodic contact with an extremely hot surface can lead to deposits in the form of crystals. This phenomenon is undesirable. Controlling the temperature at which AdBlue is injected is essential as low and very high temperatures will help avoid this phenomenon. The authors therefore propose controlling the AdBlue injection parameters according to the temperature in the SCR system. Microscopy measurements showed that deposits can approach 400 μm in just a few cycles of droplet injection, so that over time millimetre-sized plates can form at the SCR catalyst inlet surface with fundamental consequences. In conclusion the method of AdBlue injection in a controlled ultrasonic environment is effective and justified by increasing droplet break-up and avoiding crystal deposition on adjacent surfaces. 

## Figures and Tables

**Figure 1 micromachines-14-01488-f001:**
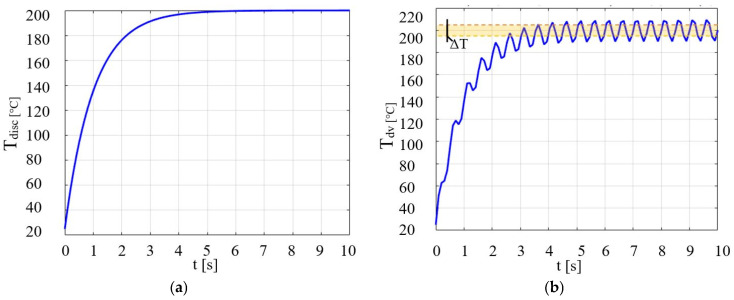
Temperature evolution of the evaporation process: (**a**) without the influence of vibration and (**b**) case with the effect of vibration.

**Figure 2 micromachines-14-01488-f002:**
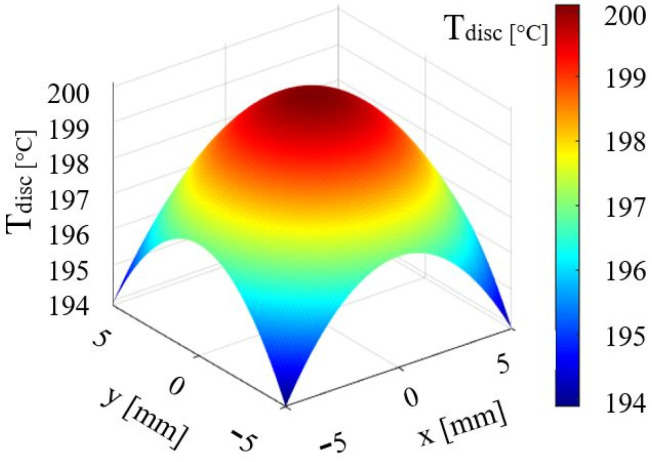
Temperature distribution on the disc surface.

**Figure 3 micromachines-14-01488-f003:**
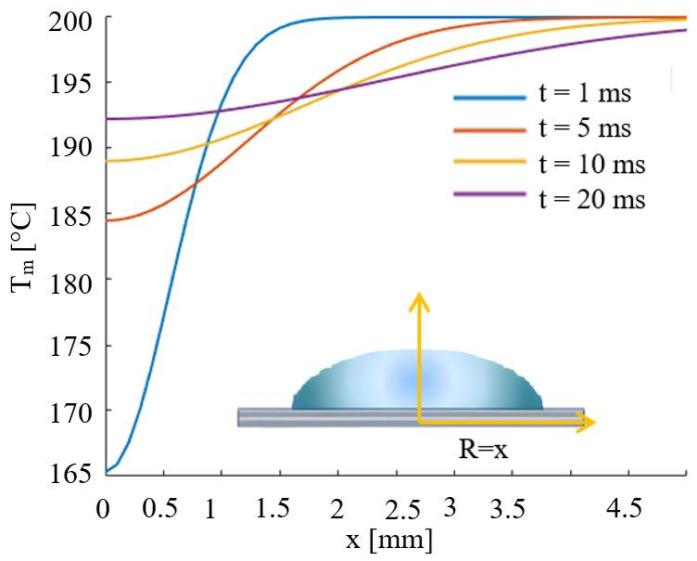
Temperature change of the disc surface at the moment of impact with an AdBlue droplet in a non-vibrating media.

**Figure 4 micromachines-14-01488-f004:**
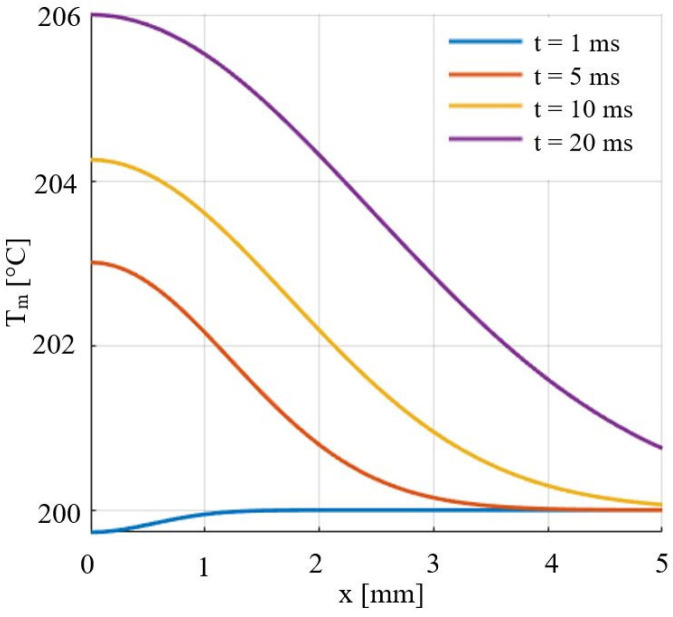
Variation of disc surface temperature at the time of impact with an AdBlue droplet in a vibrating media.

**Figure 5 micromachines-14-01488-f005:**
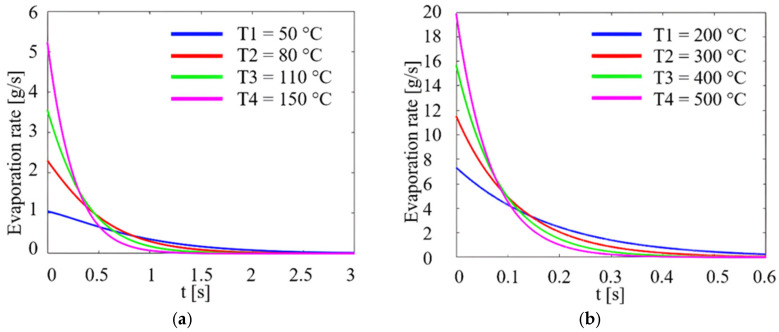
Evaporation rate of a liquid droplet as a function of time and temperature, without vibration, in a transient regime: (**a**) from temperature values from 50 °C to 150 °C, (**b**) from 200 °C to 500 °C.

**Figure 6 micromachines-14-01488-f006:**
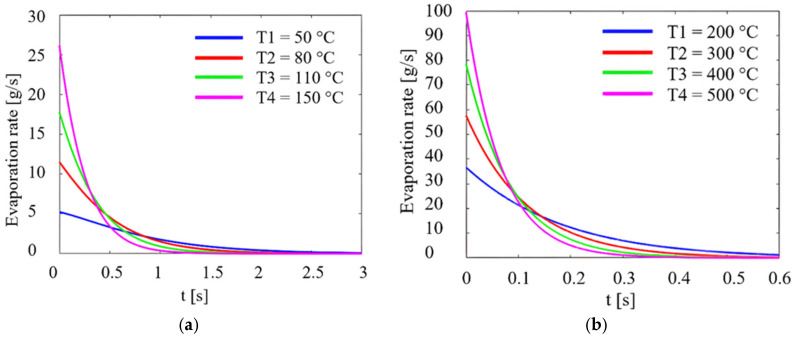
Evaporation rate of a liquid droplet as a function of time and temperature, in transient regime. (**a**) from temperature values from 50 °C to 150 °C, (**b**) from 200 °C to 500 °C, with vibration.

**Figure 7 micromachines-14-01488-f007:**
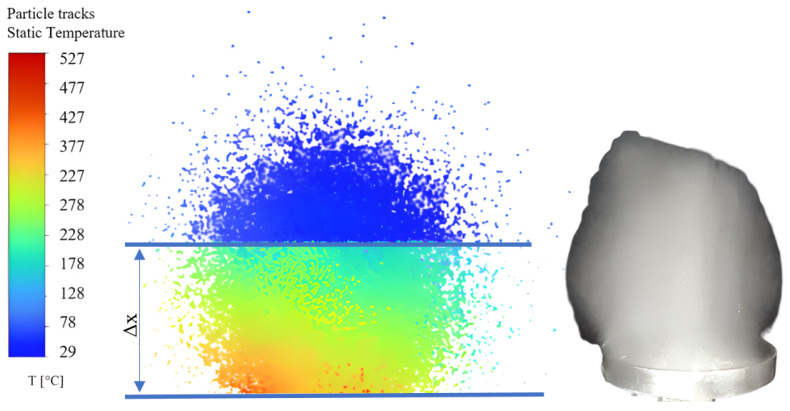
Ansys simulation of droplet evolution on impact with a vibrating disc at a frequency of 20 kHz.

**Figure 8 micromachines-14-01488-f008:**
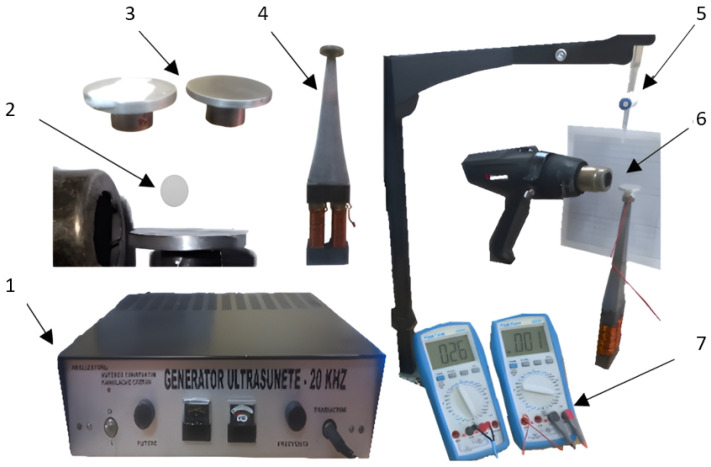
Experimental equipment for the study of AdBlue substance calefaction process.

**Figure 9 micromachines-14-01488-f009:**
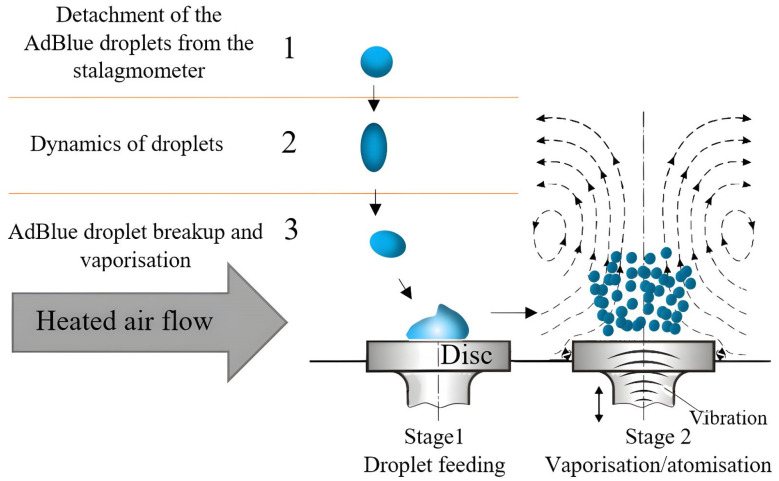
Schematic diagram of the experimental process.

**Figure 10 micromachines-14-01488-f010:**
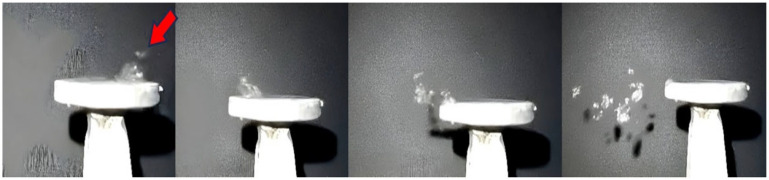
Dynamic evolution of an AdBlue drop without vibration.

**Figure 11 micromachines-14-01488-f011:**
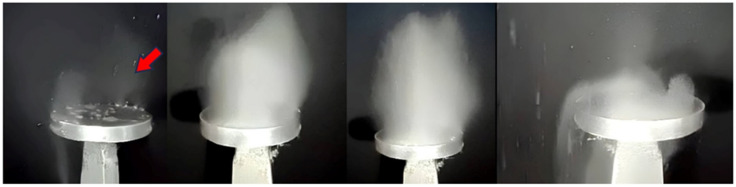
The process of calefaction of an AdBlue droplet in the presence of vibrations with a frequency of 20 kHz.

**Figure 12 micromachines-14-01488-f012:**
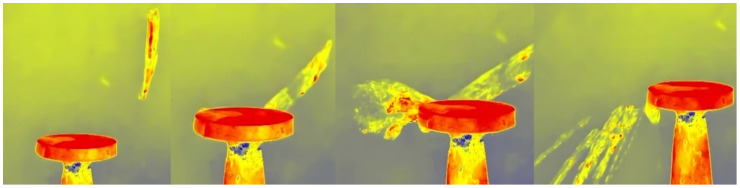
Vapour evolution for a droplet of AdBlue without vibration.

**Figure 13 micromachines-14-01488-f013:**
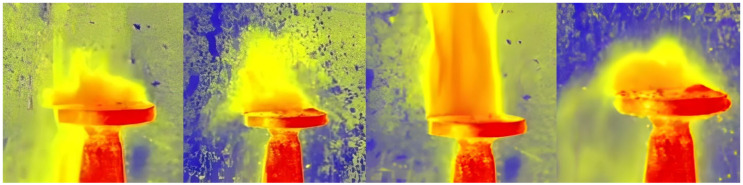
Vapour evolution of an AdBlue droplet in the condition of 20 kHz vibrations.

**Figure 14 micromachines-14-01488-f014:**
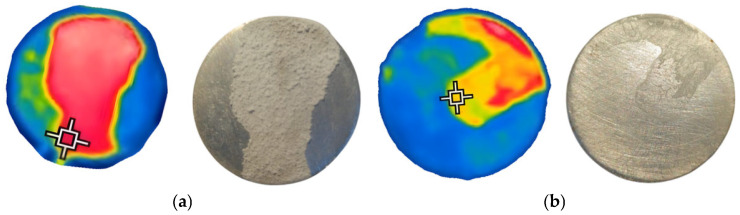
Experimental AdBlue deposition: (**a**) without vibration (**b**) with vibration.

**Figure 15 micromachines-14-01488-f015:**
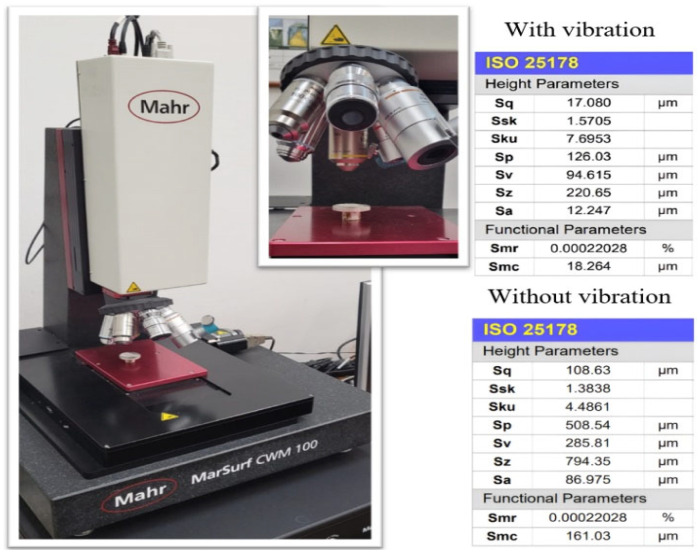
MarSurf CWM 100 confocal microscope.

**Figure 16 micromachines-14-01488-f016:**
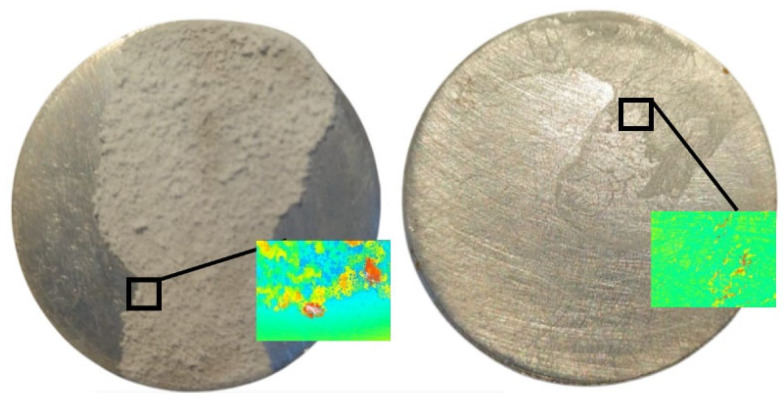
AdBlue residue deposits in absence (**left**) and presence (**right**) of vibrations, with indication of investigated areas.

**Figure 17 micromachines-14-01488-f017:**
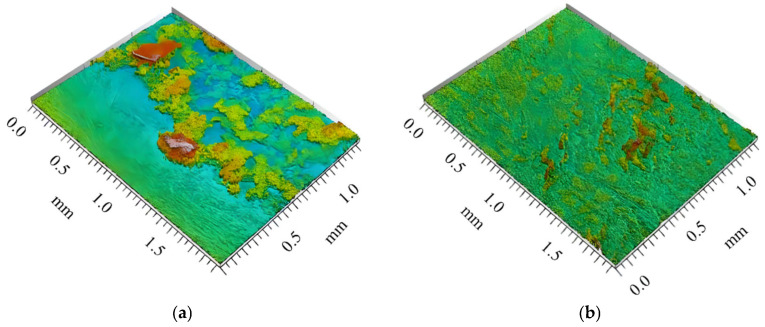
3D profiles of urea deposition in AdBlue calefaction obtained with Mahr CWM 100 confocal microscope: (**a**) without vibration, (**b**) with vibration.

**Figure 18 micromachines-14-01488-f018:**
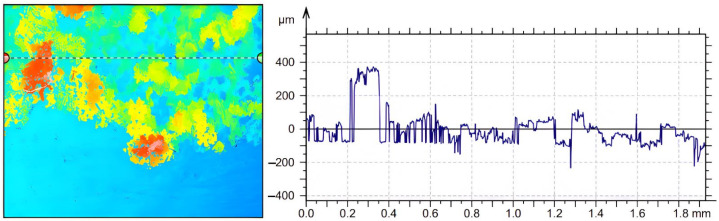
Deposition profile of AdBlue urea on calefaction in non-vibration sample.

**Figure 19 micromachines-14-01488-f019:**
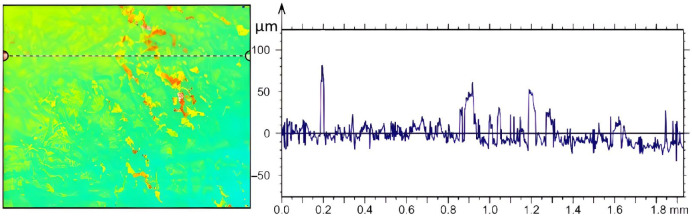
Deposition profile of AdBlue urea on calefaction in vibration test.

**Figure 20 micromachines-14-01488-f020:**
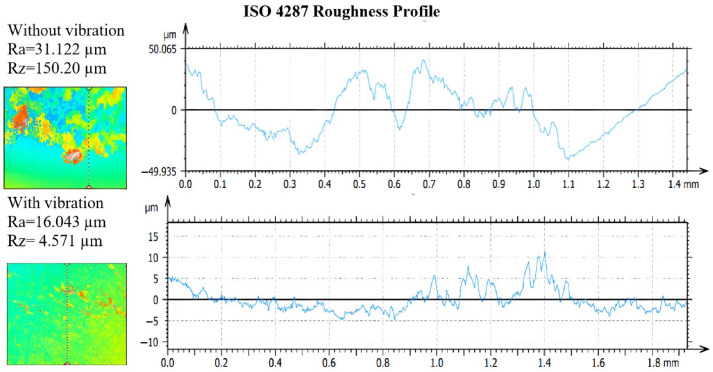
Roughness profiles of AdBlue, urea deposits by calefaction Ra and Rz.

**Table 1 micromachines-14-01488-t001:** Parameters used in the mathematical model.

Parameters	Values	Units
AdBlue droplet temperature (*T_i_*)	25	[°C]
Disc temperature (*T_disc_*)	200	[°C]
Frequency (*f*)	20,000	[Hz]

## Data Availability

Some or all data, models, or code generated or used during the study are available from the corresponding author by request.

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
