# Peer review of "Experimental Investigations upon Ultrasound Influence on Calefaction of AdBlue in Selective Catalytic Reduction Systems (SCR)"

_micromachines, 2023, doi:10.3390/mi14081488_

Round 1
Reviewer 1 Report
In the manuscript, the authors discuss the importance of eliminating all or part of the urea formations and deposits on the microchannels of SCR systems. The feasible way was put forward that injecting the AdBlue solution into an ultrasonic field. I recommend that this manuscript should be considered for publication in “Micromachines” after the following concerns are well addressed.
1. The abbreviation "SCR "shoud be defined where it first appears in the manuscript.
2. In the introduction part, the authors should give a more general and broad background of ultrasonication driven catalytic reactions and some important related papers should be cited such as Small 18 (29), 2202507; Ultrason Sonochem. 2021, 78: 105704; Nano Energy 95, 107031; Top Curr Chem (J). 2017; 375(2): 41; Angew. Chem. Int. Ed. 2019, 58, 7526.
3. Several formulas present confusing symbols.
4. Line 121, “Figures 19(a)-(b) show that there is a temperature difference for…”, Please check whether the sequence number of the figure expressed is consistent with the manuscript.
5. Whether the authors consider the contribution of energy generated by ultrasound to heat transfer.
6. Lines 356-358, What is the evidence for the formation of deposits in isocyanic acid constituent, or ruling out the formation of other by-products?
needs to be improved
Author Response
"Please see the attachment."

Reviewer 2 Report
This paper describes the effect of the ultrasound on the calefaction of Adblue in SCR system. The experimental and simulated results are new and obtained results are effective in controlling deposits formed from urea. The paper will be accepted after the following minor revisions are made.
1. P. 3, equations (4) and (5), P. 4, equations (6) and (7), P. 5 equation (8), P. 6 equations (10) and (11)
Please correct the first letter.
2. P. 5, line 152
What does “198÷200” mean? Please clarify the meaning and change the description.
3. P. 8, Figure 8
I recommend that the schematic diagram of the experimental setup is prepared.
4. P. 9 – 10, Figure 10 and Figure 11
The droplets are completely different with and without vibration. What is the mechanism that produces this difference? Please add them to the revised manuscript.
5. P. 12, Figure 15
Did you analyze the components and particle sizes of the deposits? If you did the analysis, please add the data to the revised manuscript.
- There are no major problems with English.
Author Response
"Please see the attachment."
